# Predicting Math Ability Using Working Memory, Number Sense, and Neurophysiology in Children and Adults

**DOI:** 10.3390/brainsci12050550

**Published:** 2022-04-26

**Authors:** Nienke E. R. van Bueren, Sanne H. G. van der Ven, Karin Roelofs, Roi Cohen Kadosh, Evelyn H. Kroesbergen

**Affiliations:** 1Behavioural Science Institute, Radboud University Nijmegen, 6525 GD Nijmegen, The Netherlands; sanne.vanderven@ru.nl (S.H.G.v.d.V.); k.roelofs@donders.ru.nl (K.R.); 2Wellcome Centre for Integrative Neuroimaging, Department of Experimental Psychology, University of Oxford, Oxford OX2 6GG, UK; r.cohenkadosh@surrey.ac.uk; 3Donders Institute for Brain, Cognition and Behaviour, Centre for Neuroimaging, Radboud University Nijmegen, 6525 EN Nijmegen, The Netherlands; 4School of Psychology, Faculty of Health and Medical Sciences, University of Surrey, Guildford GU2 7XH, UK

**Keywords:** mathematics, EEG, cognition, number sense, working memory, developmental neuroscience

## Abstract

Previous work has shown relations between domain-general processes, domain-specific processes, and mathematical ability. However, the underlying neurophysiological effects of mathematical ability are less clear. Recent evidence highlighted the potential role of beta oscillations in mathematical ability. Here we investigate whether domain-general (working memory) and domain-specific (number sense) processes mediate the relation between resting-state beta oscillations and mathematical ability, and how this may differ as a function of development (children vs. adults). We compared a traditional analysis method normally used in EEG studies with a more recently developed parameterization method that separates periodic from aperiodic activity. Regardless of methods chosen, we found no support for mediation of working memory and number sense, neither for children nor for adults. However, we found subtle differences between the methods. Additionally, we showed that the traditional EEG analysis method conflates periodic activity with aperiodic activity; in addition, the latter is strongly related to mathematical ability and this relation differs between children and adults. At the cognitive level, our findings do not support previous suggestions of a mediation of working memory and number sense. At the neurophysiological level our findings suggest that aperiodic, rather than periodic, activity is linked to mathematical ability as a function of development.

## 1. Introduction

Arithmetic or mathematical skills are a strong predictor of the success of one’s academic achievement, future career, and socio-economic status [1,2]. Research into the individual development of these skills is crucial for early recognition of those who develop poor mathematical skills and those who develop good mathematical skills. Present research studying the diverse working mechanisms underlying mathematical skills is mainly focused on cognitive factors as important predictors, both for children and for adults. An extensive amount of literature is available that links high mathematical ability to several domain-general processes, such as working memory and executive functions [3,4,5,6,7,8,9], and domain-specific cognitive processes, such as number sense [10,11]. Working memory is the ability to simultaneously store and process information temporarily [12], and can be separated into verbal and visuospatial components that are both related to mathematical skills [5]. Number sense is the intuitive ability to understand and handle quantities and their corresponding number symbols, i.e., non-symbolic and symbolic numerosities [13], and deficits in this ability have been linked to developmental dyscalculia [14]. It should be noted that the interrelation with mathematical ability is not well understood at the cognitive and electrophysiological level, and more research is needed to identify electrophysiological correlates that differ between poor and high mathematical skills. This study aimed to investigate a developmental framework in which working memory and number sense mediate the relation between electrophysiology and mathematical ability.

### 1.1. Age Differences in Processes Underlying Mathematical Ability

An important consideration regarding the developmental relationship between working memory, number sense, and mathematical abilities, as given by Best et al. [15], is that the strength of these relations differs in various developmental stages. For instance, number sense is less relied upon in older children [16,17,18], and there is a shift from visuospatial working memory to verbal working memory when age increases throughout primary school [19,20]. Strategy use of solving mathematical problems also changes over development [21], which are also operation dependent (i.e., between multiplication and subtraction). For example, procedural strategies are dominant early in development (i.e., high working memory burden), while fact retrieval leads to automatization of mathematical operations (i.e., less working memory burden). This shift is parallel to age-related changes in brain activity (for a detailed review see Peters and De Smedt (2018) [22]). Neuroimaging studies in both children and adults have shown that the frontoparietal network is activated during arithmetic problem solving [23,24]. Interestingly, this activation shifts from frontal to parietal areas when age increases [23]. Therefore, it should be noted that neural activations and cognitive processes that contribute to mathematical skills in the adult brain are not the same as in the developing brain [25].

### 1.2. Electrophysiological Correlates of Mathematical Ability

Electroencephalography (EEG) can help indicate which processes decline or stay stable over age and if there are any compensatory mechanisms involved in individuals with low mathematical abilities [26]. For example, frequency analysis of the EEG signal during task performance appears to be related to math performance. Frequency analysis or spectral analysis of the EEG signal provides information about the power in different frequency bands such as theta (4–8 Hz), alpha (8–13 Hz), and beta (14–30 Hz) using Fourier analysis. Synchronization of a frequency band indicates increased band power and desynchronization indicates decreased band power. Previous studies indicated that theta synchronization and lower alpha (8–10 Hz) desynchronization in the left hemisphere are associated with better mathematical performance [27,28,29,30]. This relation between theta synchronization and performance was found in children and adults, but the exact region differed between the two groups. For children, increased theta synchronization in the frontal-central and bilateral parietal-temporal areas was shown to be predictive of mathematical performance [27,30]. The authors related this increase in theta to the encoding of new information. Increased theta was found during the completion of addition and multiplication problems. For adults, theta synchronization in the left parietal-occipital areas, but not in frontal areas, was related to mathematical performance [28,29]. Lower alpha desynchronization in the bilateral parietal-occipital areas of adults is associated with subtraction and division problems solved with procedural strategies. Solving multiplication problems results in higher theta synchronization and lower alpha desynchronization, whereby less alpha desynchronization relates to increased task performance after training [27,31]. These electrophysiological activity changes have also been associated with more general cognitive processes [32,33]. For example, theta synchronization increases when working memory load increases [34,35].

Besides the importance of frequency analysis during task performance, an association between mathematical ability and frequency analysis during rest has also been found. Namely, in one recent brain stimulation study it was observed that frontal resting state activity (rs-EEG) in the beta frequency range (14–30 Hz) was linked to high mathematical performance [36]. Adults with average and high mathematical baseline skills improved more on a mathematical task when receiving stimulation in the beta range compared to stimulation in other frequency bands. It should be noted, however, that, for adults with poor mathematical skills, the greatest improvement was shown when gamma stimulation was applied. However, the relation between gamma activity (>30 Hz) and mathematical skills could not be reliably explored due to the equipment used. This study was only conducted with adults and it is unclear if the same applies to other developmental groups, e.g., children. Thus, with exception of the study from Van Bueren et al. [36], it is largely unclear if rs-EEG activity can predict mathematical skills.

### 1.3. Analyzing Beta Oscillations

Traditional rs-EEG approaches normally investigate the height of oscillatory (i.e., spectral) power in predefined frequency bands. However, these traditional analyses automatically assume the presence of oscillatory activity of the power spectral density (PSD) even if none is present (Figure 1a). In the same vein and as mentioned by Donoghue et al. [37], periodic parameters such as oscillatory power and center frequency can be conflated with aperiodic activity. Aperiodic activity is often overlooked in these traditional analyses and was previously classified as background noise in the spectrum. However, more and more evidence has arisen that shows that this activity contains inter-individual importance for behavior, and to health and disease [38,39,40,41]. This activity is the 1/f-like structure of the PSD whereby power decreases with increasing frequencies, and consists of the aperiodic offset and exponent (Figure 1a). The offset reflects the uniform shift of power across all frequencies and the exponent reflects the x parameter in the 1/fx function. It is thought that the aperiodic offset and exponent underlie neuronal spiking and that changes in the exponent relate to the excitation and inhibition (E/I) levels in the brain mediated by the neurotransmitters glutamate and GABA [42,43,44]. Overlooking aperiodic activity in EEG analyses may lead to misinterpretation. To illustrate, age can impact the center frequency and leads therefore to failures to capture spectral power in predefined band analyses such as alpha activity (8–12 Hz) [37]. For example, older participants have more lower-frequency alpha than younger participants, and therefore predefined alpha band ranges miss a proportion of power within participants. Additionally, a recent study showed that the aperiodic activity (i.e., offset and exponent) decreases with age [45]. Thus, spectral power can also be influenced by changes in the aperiodic activity.

A more sophisticated EEG analysis method has recently been established in which parameterizing neural power spectra into periodic and aperiodic components was mentioned [37]. Since the relation between rs-beta activity and mathematical ability was found using the traditional method and due to the aforementioned criticism on this method, we compared this method with the new parameterization method using the open-source Python algorithm Fitting Oscillations and One-Over-F (FOOOF) from Donoghue et al. [37] (see Figure 1b for an illustration).

### 1.4. The Present Study

This study was inspired by our recent finding from Van Bueren et al. [36] that rs-beta activity may be an important neural correlate of mathematical skills. Altogether, we studied a developmental framework of mathematical skills from brain to behavior. This framework entails the interrelation between neural characteristics (rs-beta activity), domain-general (working memory), and domain-specific (number sense) cognitive factors, and behavior (mathematical skills) between children and adults.

First, we investigated if there is a relation between electrophysiological activity during rest and mathematical ability that is moderated by age and mediated by working memory and number sense. Figure 2a shows our moderated mediation model with our corresponding hypotheses. Next, Figure 2b shows the statistical model of this conceptual model. Firstly, we expect that higher frontal rs-EEG power in the beta frequency range relates to higher mathematical ability (path C’), and to higher working memory and number sense scores (path A). Note that we focus on path C and not path C’ (i.e., relation between rs-EEG and mathematical ability) to match the results of Van Bueren et al. [36]. Furthermore, it is expected that high achievers in mathematical ability also display higher performance in both working memory and number sense (path B). We standardized beta activity, working memory and number sense (path D), and mathematical ability (path E) based on age group (children vs. adults) since we are not interested in the mean difference but in the interrelations. Therefore, the coefficients of path D and E will be zero and not further mentioned. Path C_mod_ is the core of this study, and entails the moderation of age group of the relation between frontal beta rs-EEG and mathematical ability. Since, to the best of our knowledge, there are no studies known that investigated the relation between frontal beta rs-EEG activity and mathematical abilities, with the exception of the study from Van Bueren et al. [36] that tested adults, we investigated if age group moderates the relation between beta rs-activity and mathematical ability. As found in Van Bueren et al. [36], we expect that adults who have higher beta activity also have higher mathematical abilities (C_mod_). However, we do not have an expectation regarding the moderation effect of age group between rs-beta activity and working memory/number sense (path A_mod_). In addition, we expect that children and adults who have a strong number sense [46] and working memory [47] also have stronger mathematical abilities (path B), and that this effect is stronger for children (path B_mod_).

These findings may contribute to developing an intervention at a neuroscientific level (i.e., brain stimulation targeting beta oscillations). We expect that the more recent and proficient parameterization method of calculating rs-beta activity has more predictive power in the proposed framework compared to the traditional method of calculating rs-beta activity. Recent research showed that aperiodic components (offset and exponent) differ with age and are related to behavior [38,41,45], and we therefore explored their relation with mathematical ability. As mentioned in our preregistration on the Open Science Framework (see osf.io/37pyn), we conducted exploratory analyses to look at resting state theta and alpha activity calculated with the parameterization method, in addition to beta activity. We looked at frontal-central, bilateral parietal-temporal, left parietal-occipital, and bilateral parietal-occipital areas based on previous studies that used the traditional method assessing EEG during task performance [27,28,30,31].

## 2. Materials and Methods

### 2.1. Participants and Ethics Statement

In total, 52 healthy, right-handed children aged 9–10 and 58 adults aged 18–35 were recruited around Nijmegen. Participants were excluded if they were pregnant or thought they were, and if they had any personal or family history of neurological and psychiatric disorder including dyscalculia and ADHD. Two participants (1 adult and 1 child) were excluded due to technical malfunction and two participants (1 adult and 1 child) were excluded due to noisy EEG data. Additionally, data of both the Odd One Out task and non-symbolic comparison task from 1 child were not saved, and these tasks for this participant were therefore excluded. The final sample consisted of 50 children aged 9 or 10 years old (M = 9 years, 7 months, SD = 0.64), and 56 adults (M = 23.28 year, SD = 2.76). Participants received financial compensation for participation (10 euros per hour in vouchers) and travel expenses if applicable. All participants were naïve to the aim of the study and written informed consent was obtained before the start of the experiment from either the participant themselves or a parent. The present study is in compliance with the standards set by the Declaration of Helsinki and ethical approval was obtained by the ethical advisory committee of the Radboud University (ETC-GW number ECSS-S-18-00240, 22 February 2019).

### 2.2. Overview of Experimental Paradigm and Stimuli

Before explaining the tasks in more detail, we first present a schematic overview of the experimental paradigm. As can be seen in Figure 3, an rs-recording of four minutes was made at the beginning of the experiment to measure beta activity during rest. After removal of the EEG-cap and a short break of around 3 min, a Dutch speeded arithmetic Test (Tempo Toets Rekenen, TTR; Lisse, The Netherlands [48]) was administered, in which participants had to solve as many arithmetic problems as they could within 5 min in total. After completing this test, number sense was measured by means of a computerized test battery consisting of a number line estimation task and a non-symbolic and a symbolic comparison task (~15 min) [49,50]. Lastly, both verbal and visuospatial working memory were assessed with the computerized Backward Digit Recall task and the Odd One Out task (~10 min) based on the Automated Working Memory Assessment battery [51]. In total, participants completed the experiment within 2 h including EEG preparation. Data collection was performed by the first author, who was trained in conducting research with the electroencephalogram, or by trained graduate students.

#### 2.2.1. Resting-State Electroencephalography Recordings and Pre-Processing

Rs-EEG recordings of four minutes were made before the start of the experiment. Data were obtained with 32 Ag/AgCl electrodes according to the international 10/20 EEG system using the Biosemi ActiveTwo system at 500 Hz with no online filters (Biosemi, Amsterdam, The Netherlands). Additional electrodes were placed below both eyes and next to the right eye to record eye blinks. Impedances of the electrodes were held below 5 kΩ. The ground consisted of the active common mode sense (CMS) and passive driven right leg (DRL) electrode, which was positioned on the left mastoid. During the recording, all participants were instructed to watch a fixation point in the middle of a computer screen with eyes opened, and to try to avoid mental and muscular activity. Pre-processing was undertaken using the open source toolbox EEGLAB (v13.6.5.b; [52]) that runs in MATLAB (R2020b). First, filtering was performed by applying a high-pass filter of 0.1 Hz to minimize slow drift to raw EEG data, and a Notch filter to remove line noise. Next, we manually checked every EEG data file and removed any high frequency artifacts due to muscle movement. EEG data files were rejected if more than 25 percent of the files needed to be rejected (2 participants, as explained in the Section 2.1). Subsequently, we performed an Independent Component Analysis (ICA) to remove stereotyped artifacts such as eye movements, blinks, heart rate activity, and muscular activity. This resulted in an average of 1.21 ± 0.54 SD components rejected per participant with a maximum of 3 components and a minimum of zero.

#### 2.2.2. Arithmetic Ability

A Dutch speeded arithmetic Test, Tempo Toets Rekenen (TTR; [48]) was assessed on paper to measure individual arithmetic ability. Participants were instructed to solve as many arithmetic single- and double-digit operations (additions, subtractions, multiplications, divisions, and mixed) within 1 min for each subtest of operation. Participants were also instructed to solve the problems from the top of the page to the bottom of the page. Each correct item of each subtest was awarded one point. We used the standardized sum-score of every subtest in the main analysis.

#### 2.2.3. Cognitive Measurements

##### Number Sense

An adjusted computerized number line estimation task of the computerized Dutch Assessment Battery for Number Sense (DANS) was used [49]. The number line estimation task ranged from 0 to 1000 and included a short training of 2 trials at the beginning and 42 test trials in total. The final score entailed an error score that was the average deviation of all trials, in which a lower score indicated higher performance. In other words, the mean absolute error (MAE) was computed by averaging the absolute difference between the targets and their estimated positions. Additionally, two other subtasks of the computerized DANS were used to assess non-symbolic and symbolic number sense: the non-symbolic comparison task included 6 training trials and 42 test trials, and the symbolic comparison task included 6 training trials and 33 test trials with variable difficulty. The goal of both comparison tasks (that ranged from 0 to 100) was to indicate with a button press which of two sets of dots (non-symbolic) or numbers (symbolic) that were presented on a screen was the largest. The number and size of the dots differed per trial and the maximum allowed response time for both tasks was 5000 ms. For the non-symbolic task, we analyzed the accuracy. For the symbolic task, we analyzed response times (RTs) for the correct trials due to expected ceiling effects in the adult group. Note that responses below 200 ms, training, and incorrect responses were removed before analysis. Then, responses from the number line estimation task and the symbolic comparison task were mirrored in such a way that higher scores resemble higher performance. This was performed by calculating the difference between the maximum score (i.e., 1000 for the number line estimation task and 5000 ms for the symbolic comparison task) and the performance score. Finally, a mean score was calculated after standardizing the number line estimation task score and both comparison tasks outcomes to provide a composite score of number sense per participant. To determine the strength between the different scores combined into this composite score, we ran a Pearson’s correlation (all *p* < 0.045, all r > 0.20).

##### Working Memory

Lastly, visuospatial working memory and verbal working memory were measured with two computerized tasks based on the Automated Working Memory Assessment (AWMA) battery [51]. Specifically, verbal working memory was measured with a Backward Digit Recall task and visuospatial working memory with an Odd One Out task. Both tasks started with a short training, and the participant moved to the next block (which ranged from 2 to 7) if the participant recalled four trials within a block correctly, and the task was discontinued when the participant made three mistakes within one block. During the Backward Digit Recall task, participants had to recall a number sequence in reverse order (that ranged from 2 to 7) that was verbally presented. A higher accuracy score indicated a higher verbal memory. During the Odd One Out task, participants had to point to the stimulus that was the odd one out (out of three). Then, three empty boxes appeared and the participant had to recall the location of the previous stimulus (left, middle, or right). Note that this task also ranged from a block with a sequence of 2 to a sequence of 7. A higher accuracy score on this task indicated a higher visuospatial memory. We computed a mean score of overall working memory by averaging both verbal and visuospatial working memory standardized outcomes. We ran a Pearson’s correlation to determine the strength between verbal and visuospatial working memory (*p* < 0.001, r > 0.47).

### 2.3. EEG Analysis

#### Resting-State Beta Activity (Traditional and Parameterization)

Traditionally, the amount of rs-beta activity is calculated by transforming the data into the frequency domain via Fast Fourier Transformation (FFT) using a window length of 2000 data points with an overlap of 1000 [36]. Beta activity in the 14–30 Hz range is calculated according to the relative power (μV^2^), and normalized by means of dividing the absolute frequency power of the frequency band by the average absolute power in the 4–30 Hz range. Note that the relative power is the ratio of the frequency band of interest (i.e., beta) to the total frequency band (i.e., theta to beta; 4–30 Hz). The more recent parameterization method from Donoghue et al. [37] was used to extract individual periodic beta activity (i.e., power in the beta peak) after removing aperiodic activity with the Fitting Oscillations and One-Over-F (FOOOF) Python algorithm v1.0.0. Settings contained a maximum of 5 peaks to reduce overfitting, a peak width limit of 1–8 Hz and no knee was fitted to the data. Note that we extracted the power of the highest peak within the beta frequency range (over and above the aperiodic component), and if no peak was found, a value of zero was returned. Thus, we used two methods to calculate rs-beta activity (the traditional and parameterization methods), which means that we ran our moderated mediated analysis twice, with each method separately.

### 2.4. Data Analysis

The present study was pre-registered on the Open Science Framework, see osf.io/37pyn. We ran two moderation analyses (Model 1) and two moderated mediation analyses (Model 59) using the Hayes’ user defined function PROCESS in R (version 4.0.1) with the total TTR scores (arithmetic ability) as dependent variable. Beta activity was the independent variable (calculated with either the traditional or parameterization method), and age group (children vs. adults) was the moderator. Additionally for Hayes’ Model 59, working memory (M1) and number sense (M2) were included as parallel mediators. We used 10,000 bootstrap samples, and determined 95% confidence intervals (CI) for direct and indirect effects. If the CI intervals do not include zero, the effect is considered significant. We z-transformed all continuous predictors based on age before running the analysis. We report the coefficients (β) and the standard error (SE) of all model paths.

Additionally, we compared the parameterization method from FOOOF with the more traditional method, as used in Van Bueren et al. [36], to see how both analysis methods differ in measuring beta activity and their relation to our behavioral outcomes. In other words, we compared the output using FOOOF with the relative frequency band power by looking at the correlation coefficient with aperiodic activity. To do this, we also computed the aperiodic activity in the beta frequency range with the FOOOF package and compared it with the traditional analysis method to check if both methods largely measured the same activity. For the exploratory analyses, we ran independent sample *t*-tests to compare the aperiodic offset and exponent between children and adults. We also explored the relation between aperiodic activity (average score of the offset and exponent; independent variable) and mathematical ability (i.e., dependent variable) by means of a simple moderation (Hayes’ Model 1) with age group as moderator. As stated in our preregistration, we also exploratorily computed the theta (4–8 Hz) and alpha (8–12 Hz) peak power calculated with the parameterization method. This was done for the frontal-central area (electrodes Fz and FCz), the bilateral parietal-temporal area (electrodes T7, T7, T8, and P8), the left parietal-occipital area (electrodes PO9, P3, and O1), and the bilateral parietal-occipital area (electrodes PO9, P3, O1, PO10, P4, and O2). We ran these additional moderated mediation models with different dependent variables: one using the total TTR scores, the sum TTR scores for addition and multiplication problems (i.e., predominantly retrieval strategies), and one using the sum TTR scores for subtraction and division problems (i.e., predominantly procedural strategies).

## 3. Results

Table 1 provides the descriptive statistics.

### 3.1. Moderated Mediation Models: The Parameterization Method and the Traditional Method

First, we wanted to determine if frontal rs-beta activity is related to mathematical ability and if this relation differs with age. Therefore, we first present the results of a simple moderation analysis with total TTR score as dependent variable and beta activity as independent variable moderated by age group (Hayes’ Model 1). Then, we wanted to determine if working memory and number sense are important mediators. Hence, we subsequently present the outcomes of the moderated mediation models in Figure 2 with working memory (1) and number sense (2) as parallel mediators (Hayes’ Model 59). The outcome variable entails the total TTR score, i.e., math ability, and age group (child or adult) was included as moderator. The independent variable is the beta periodic power as calculated with the recent parametrization method (i.e., FOOOF) as shown in Figure 4a and the relative power of the traditional method as shown in Figure 4b.

We determined the strength of the effect between beta activity and mathematical ability by running a model that only included age as moderator but no mediators (Hayes’ Model 1). For the traditional method, path C_mod_ is significant (path C_mod_; *β* = −0.40, *SE* = 0.19, *p* = 0.037). However, when inspecting the simple slopes, there was no relation between beta activity and mathematical ability for children (path C; *β* = 0.22, *SE* = 0.14, *p* = 0.11) and adults (path C; *β* = −0.18, *SE* = 0.13, *p* = 0.17), indicating no difference. Similar results for the simple slopes were found for the parametrization method for path C_mod_ (*β* = 0.06, *SE* = 0.19, *p* = 0.75) and C (*β* = −0.15, *SE* = 0.14, *p* = 0.27). In short, no relation exists between frontal rs-EEG and mathematical ability, and this is not different for children and adults.

To determine if working memory and number sense mediate the relation between beta activity and mathematical ability, we inspected Hayes’ Model 59 that includes parallel mediators. The significant results that were found relate to path B of both models (see Figure 4). Working memory is strongly related to mathematical ability independent of age. To elaborate, children and adults who show stronger mathematical abilities also perform better in the working memory tasks. No relation was found between number sense and mathematical ability, either in children or in adults. It should be noted that the results indicate a weak (i.e., trending) relation between rs-beta activity and working memory and number sense (path A), when analyzed with the traditional method, but not with the parameterization method. This shows that both analysis methods may indeed lead to slightly different outcomes in relation to cognitive and behavioral measurements.

### 3.2. Comparisons of the Spectral Calculation Methods

We compared the parameterization method with the more traditional method as normally used in spectral analysis to see how both differ in capturing periodic activity in the beta frequency range. We compared both methods by investigating the correlation coefficients with aperiodic activity in the beta range as shown in Table 2. 

Firstly, the results as presented in the correlation matrix in Table 2 show no significant correlation between the traditional calculation method and the new parametrization method. This shows that there is no relation between the measured power as calculated with the two different EEG analyzing methods, indicating that both methods are largely assessing different aspects of the PSD. The correlation coefficient between beta activity calculated with the traditional method indicates a negative moderately sized relation with aperiodic activity in the beta frequency range. Conversely, the parameterization method indicates a weak positive relation to aperiodic activity. In short, these findings indicate that the traditional method is not just calculating spectral beta activity but also entails aperiodic activity. This relation is stronger when compared to beta activity calculated with the parameterization method. This result strengthens the notion that traditional EEG analyses conflate spectral power analyses with aperiodic activity [37].

### 3.3. Exploratory Analyses

When running an independent samples *t*-test to compare the aperiodic components between age groups, we found a significant lower offset in adults than in children, similarly to Cellier et al. [45] (see Table 1). In contrast to their results, we found that the exponent was significantly smaller in adults than in children, as shown in Table 1. Since the working mechanisms of aperiodic activity have been related to E/I levels of the brain, which in turn relates to mathematical achievement [53], we exploratorily looked at the interaction between aperiodic activity and age on mathematical ability. Interestingly, a simple moderation (Hayes’ Model 1) with aperiodic activity (average score of the exponent and offset) as independent variable showed a significant interaction with age in predicting mathematical ability (*β* = 0.53, *SE* = 0.19, *p* = 0.006). Reflecting an increased excitation/inhibition ratio, children with low aperiodic activity have higher mathematical abilities (*β* = −0.36, *SE* = 0.13, *p* = 0.009), whereas no relation was found in adults (*β* = 0.16, *SE* = 0.19, *p* = 0.21). To conclude, aperiodic activity predicted mathematical ability in children. When running the full model (Hayes’ Model 59) for aperiodic activity with working memory and number sense as parallel mediators, no mediation effects were found.

In addition to beta activity, we exploratorily investigated theta and alpha peak activity in the frontal-central (electrodes Fz and FCz), bilateral parietal-temporal (electrodes T7, T8, and P8), left parietal-occipital (electrodes PO7, PO3, and O1), and bilateral parietal-occipital areas (electrodes PO9, P3, O1, P4, P8, and O2) calculated with the parameterization method based on previous literature [27,28,30,31]. Results of these moderated mediation models showed no association with mathematical ability, nor a moderation by age group (for a detailed overview of the output see the Appendix A).

Our results do not show a relation between number sense and mathematical ability, contrary to previous studies [7,10,11]. Moreover, the subscores of the individual number sense tasks (number line estimation task, symbolic and non-symbolic tasks) were weakly related to each other (all *p* < 0.045, all r > 0.20). Therefore, a new moderated mediation model was run with the individual number sense subscores included as parallel mediators (Hayes’s Model 59 including working memory as mediator). This was undertaken to determine whether a specific number sense skill (number line estimation, symbolic and non-symbolic comparisons) was related to mathematical ability. Number line estimation did not predict mathematical ability (*β* = 0.17, *SE* = 0.11, *p* = 0.12), nor did symbolic comparison (*β* = 0.09, *SE* = 0.09, *p* = 0.35) or non-symbolic comparison (*β* = 0.14, *SE* = 0.12, *p* = 0.24). Additionally, rs-beta activity calculated with the traditional method did not predict number line estimation (*β* = 0.24, *SE* = 0.13, *p* = 0.06), symbolic comparison (*β* = 0.00, *SE* = 0.13, *p* = 0.96), or non-symbolic comparison (*β* = 0.04, *SE* = 0.14, *p* = 0.75). For beta activity calculated with the parameterization method a significant relation was found for number line estimation (*β* = −0.32, *SE* = 0.12, *p* = 0.012), but not for symbolic comparison (*β* = 0.11, *SE* = 0.12, *p* = 0.38), or non-symbolic comparison (*β* = −0.13, *SE* = 0.14, *p* = 0.34). These results indicate no relation between number sense and mathematical ability for different age groups, nor a relation with resting-state electrophysiology.

## 4. Discussion

We investigated a developmental framework of mathematical skills from brain to behavior, with an interrelation between neural (beta oscillations during resting state), cognitive (working memory and number sense), and behavioral (mathematical ability) factors. Contrary to our expectations, we found no relation between resting state electrophysiology and mathematical ability for children and adults. Moreover, no mediation was found by working memory and number sense. We did find that higher performance in working memory tasks, but not number sense, was related to higher mathematical abilities. By comparing both EEG analysis methods focusing on the beta frequency range, we found that the results of the traditional method show moderate overlap with aperiodic activity, indicating that this method does not solely measure periodic activity. Moreover, we found that aperiodic components (i.e., the exponent and offset) were lower for adults compared to children. Since it is thought that this activity is related to neuronal spiking and E/I levels, we exploratorily investigated if aperiodic activity predicted mathematical ability. This was indeed found and this relation was moderated by age group, in which children with low aperiodic activity have high mathematical abilities whereas no relation was found in adults.

Although we expected a mediation effect, because previous studies showed that both domain-general and domain-specific cognitive factors such as working memory and number sense are strong predictors of mathematical abilities [3,4,5,6,7,8,9,10,11], the results did not show such an effect. In line with previous behavioral research, we did find a strong relation between working memory and mathematical abilities, but not for number sense. This can be explained due to the finding that number sense in older children and adults is less relied upon when solving mathematical problems [16,17]. We tested relatively older children aged 9 and 10 who may not rely on number sense for solving mathematical problems. Moreover, the subscores (individual number sense tasks) of the composite score for number sense were weakly related to each other, which might have impacted the results. To investigate this further, we tested a new moderated mediation model in which the three number sense scores were included as separate mediators together with working memory. In this model, the three separate number sense scores did not predict mathematical ability, and this relation was not mediated by age. Moreover, rs-beta activity predicted neither of the three number sense mediators when beta activity was determined with the traditional parameterization model. Only rs-beta activity calculated with the parameterization method was a negative predictor of number line estimation. However, since number line estimation was only weakly related to other number sense task scores (i.e., symbolic and non-symbolic comparison), we found no strong evidence in favor of an overall relation between number sense and mathematical abilities, or between number sense and resting-state electrophysiology. In short, children and adults who score high on working memory tasks, also have higher mathematical abilities independent of resting state electrophysiology.

In the present study, we especially focused on the predictive value of frontal rs-beta activity. This is in contrast to previous mathematical studies that mainly focused on frequency analysis during task performance. These EEG studies indicated the importance of theta synchronization and alpha desynchronization during a task that involved solving mathematical problems. Nevertheless, a recent study linked frontal rs-EEG activity in the beta frequency range (14–30 Hz) to high mathematical performance [36], as confirmed by electrical brain stimulation results. However, this study only included adults and no other developmental groups. When running a simpler model with age group as moderator but no mediators, we found a weak trending interaction between frontal rs-beta activity and mathematical ability (path C_mod_), but only using the traditional frequency analysis. When running the same simplistic model with this parameterization method, we found no relation between frontal rs-beta activity and mathematical ability moderated by age. This is noteworthy, since this supports the idea that the spectral power can be conflated with aperiodic activity, and overlooking this may lead to misinterpretation [37]. In our case, a wrong conclusion (i.e., that periodic activity is related to mathematical ability) could have been drawn based on the trending effect when ignoring the results of the parameterization method. Moreover, the two analysis methods also slightly differ in their relation to behavioral and cognitive outcomes, which may be explained by the relation/conflation with aperiodic activity. Our correlation coefficient comparisons indicate a moderately negative relation between the traditional method and aperiodic activity in the beta frequency range, hinting towards shared variance. This relation was weaker for the parametrization method, which confirms a better separation between periodic and aperiodic signals. Since a recent study indicated that aperiodic activity decreases over age [45], we ran an exploratory analysis that confirmed that aperiodic activity (i.e., offset and exponent) was lower in adults than in children. Note that Cellier et al. [45] found no relation between age and the aperiodic slope (i.e., exponent) in the frontal-midline electrode cluster. This difference may be explained by a change in electrode clustering compared to this study. Although Cellier et al. [45] clustered electrode Fz with its surrounding electrodes (E4, E5, E11, E12, E16, E18, and E19), we used Fz only. These results indicate that age is an important factor that needs to be considered in electrophysiology analyses, especially when considering aperiodic activity. Therefore, we substituted rs-beta activity with aperiodic activity in the moderation model and found a relation between aperiodic activity with mathematical ability moderated by age group, but only for children. This interaction showed stronger mathematical abilities for children with low aperiodic activity.

The thought that aperiodic activities underlie neuronal spiking, which relates to the E/I levels of the brain, matches with the notion that beta oscillations reflect inhibitory GABA currents in human EEG. Animal studies show that GABA activity is related to beta oscillations [54,55]. This is backed-up by human studies related to children who have duplications of the GABAA receptor subunit genes and therefore show spontaneous beta oscillations [56], and by children with deletions of GABAA who show reduced beta oscillations [57]. These studies confirm that increased oscillations in the beta frequency range during resting state relate to the lower E/I ratio of the brain. However, aperiodic activity is not considered in these studies. The possibility exists that the periodic measures are conflated with the aperiodic ones and that beta oscillations are not underlying inhibition, but that aperiodic signals play an important role. This explanation may explain the correlation found between frontal rs-beta activity and mathematical skills in Van Bueren et al. [36]. The traditional analyzing method may have partially captured aperiodic activity, which is thought to relate to the E/I in the brain, which subsequently underlies learning [53]. The magnetic resonance spectroscopy (MRS) study from Zacharopoulos et al. [53] found that decreased E/I leads to enhanced mathematical skill in the developing brain, and that the opposite happens in the adult brain. Note that our study found exactly the opposite as Zacharopoulos and colleagues [53] for children, which may be due to MRS and EEG differences that capture different aspects of E/I. A recent finding from our lab supports this notion by showing that mathematical learning can influence the aperiodic exponent (i.e., E/I) and that an excitatory form of neurostimulation can influence this directly [58]. More evidence in favor of the idea that aperiodic activity (E/I level) is important in linking resting-state electrophysiology to mathematical skills, are the results regarding our exploratory analyses, which were preregistered. Based on previous literature using EEG during mathematical task performance, we investigated theta and alpha periodic power as calculated with the recent parameterization analysis. No effects were found for the frontal-central, bilateral parietal-temporal, left parietal-occipital, and bilateral parietal-occipital areas in relation to mathematical abilities as a function of age (see Appendix A).

In summary, the present study indicates the importance of parameterization of periodic and aperiodic electrophysiological activity when investigating the relation with mathematical skills over age. No strong evidence was found that periodic activity in the theta, alpha, and beta band predicts mathematical skills in children and adults. However, aperiodic activity of the power spectrum (i.e., E/I of the brain) is a better indicator of mathematical skills, and is moderated by age. It is likely that this not only translates to the mathematical domain, but that this finding is also an important consideration for other behavioral domains investigating neural correlates.

## Figures and Tables

**Figure 1 brainsci-12-00550-f001:**
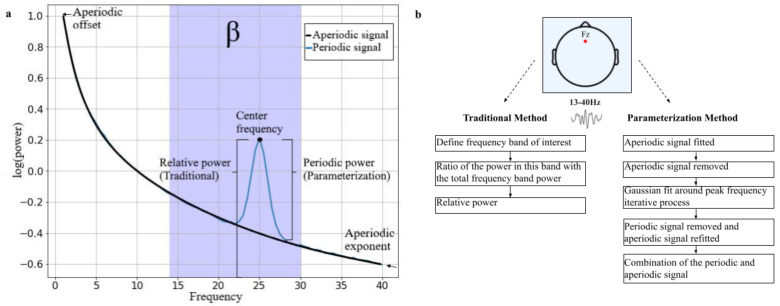
**Theoretical power spectrum and overview of differences between EEG analyses.** (**a**) PSD with the aperiodic signal (offset and exponent) indicated in black and the periodic signal indicated in blue. The shaded blue area is the beta frequency range (14–30 Hz). The relative power as calculated with the traditional method is not considering aperiodic signals while the parameterization analysis is (i.e., periodic power). (**b**) The traditional method as discussed in this paper focuses on relative power in a predefined frequency band and the parameterization method distinguishes between periodic and aperiodic signals.

**Figure 2 brainsci-12-00550-f002:**
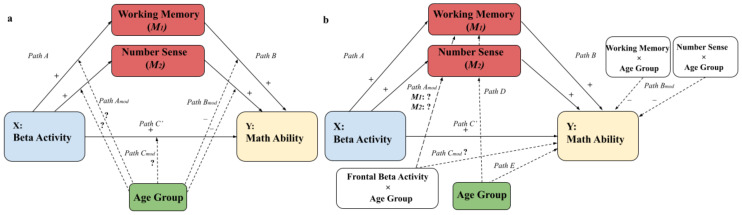
**Visualization of the moderated mediation model.** (**a**) Conceptual model with the expected direction of the hypotheses shown in the separate paths. (**b**) Statistical model with the expected direction of the hypotheses. Dashed arrows indicate moderation effects.

**Figure 3 brainsci-12-00550-f003:**
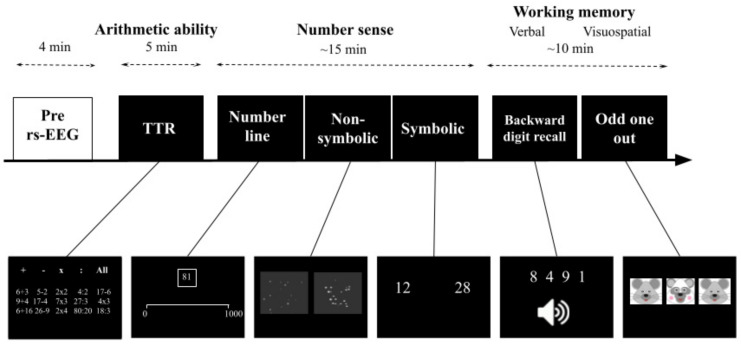
**An overview of the experimental paradigm.** All participants (*n* = 105) first completed a 4 min rs-EEG. Next, the participant completed the TTR which consisted of 5 different subtests, which took 5 min in total. Subsequently, number sense was assessed with three computerized tasks (a number line estimation task and two comparison tasks), which had a duration of around 15 min. Lastly, verbal and visuospatial working memory were assessed with the Backward Digit Recall task and the Odd One Out task, which lasted around 10 min.

**Figure 4 brainsci-12-00550-f004:**
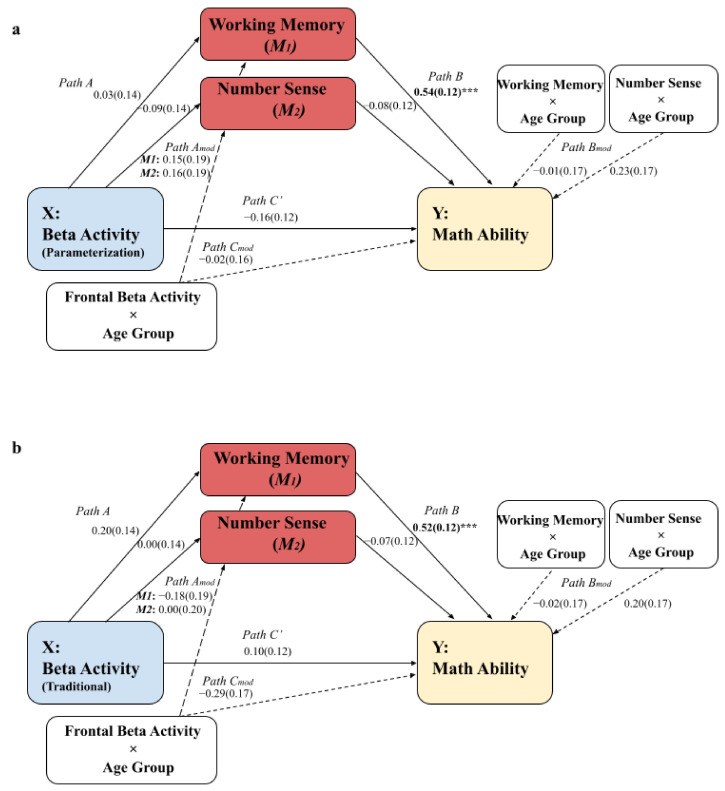
**Moderated mediation models (*n* = 105).** (**a**) Moderated mediation model that includes the frontal rs-beta activity calculated with the parameterization method. (**b**) Moderated mediation model that includes the frontal rs-beta activity calculated with the traditional method. Dashed arrows indicate moderation effects. *** *p* < 0.001.

**Table 1 brainsci-12-00550-t001:** Descriptive statistics for children and adults separately.

	Children		Adults		Mean Comparison	
Variables	Mean	SD	Mean	SD	*t*-Value	*p*
TTR Total	100.84	28.84	147.83	25.33	−8.86	<0.001
Error score NLE	44.73	22.76	28.22	10.54	4.69	<0.001
RTs Symbolic	1125.09	650.88	648.63	82.24	5.13	<0.001
Accuracy Non−Symbolic	29.77	3.41	30.35	2.90	−0.93	0.35
Accuracy BDR	10.78	3.33	14.76	4.09	−5.52	<0.001
Accuracy OOO	15.46	3.69	20.41	2.98	−7.46	<0.001
Number Sense (combined)	1617.72	219.22	1784.49	28.72	−5.28	<0.001
Working Memory (combined)	13.13	2.86	17.58	2.81	−8.02	<0.001
Beta parameterization	0.27	0.17	0.26	0.16	−0.11	0.90
Beta traditional	0.39	0.17	0.46	0.15	−2.13	0.035
Aperiodic activity ^1^ (13–40 Hz)	1.33	1.20	0.69	1.18	2.79	*p* < 0.001
Aperiodic activity ^1^ (1–40 Hz)	0.90	0.29	0.47	0.28	7.64	<0.001
Offset	0.37	0.29	−0.23	0.30	10.67	<0.001
Exponent	1.43	0.31	1.19	0.29	4.07	<0.001

TTR, Speeded Arithmetic Test; NLE, Number Line Estimation; BDS, Backward Digit Recall; OOO, Odd One Out. ^1^ Note that the aperiodic activity is the average score of the offset and exponent.

**Table 2 brainsci-12-00550-t002:** Correlation matrix between the EEG analyses methods and aperiodic activity (*n* = 106).

	Beta Parameterization	Beta Traditional
Beta Parameterization	-	
Beta Traditional	0.05	-
Aperiodic Activity	0.35 ***	−0.54 ***

The Pearson’s correlation coefficient is stated with *p*-value between brackets. *** *p* < 0.001.

## Data Availability

All datasets generated and analyzed are available on the Open Science Framework (see osf.io/37pyn), together with code used in the present study.

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
