# Peer review of "Predicting Math Ability Using Working Memory, Number Sense, and Neurophysiology in Children and Adults"

_brainsci, 2022, doi:10.3390/brainsci12050550_

Round 1
Reviewer 1 Report
In the current study, the authors investigated the role of working memory and number sense abilities in mediating the relationship between mathematical ability and its electrophysiological correlates (rs-EEG) by using a moderated mediation model in children and adults. The authors compared a traditional EEG analysis method with a novel parameterization method considering both periodical and aperiodic activities. Results showed that WM ability, but not number sense, predicts mathematical skills. When the aperiodic components were considered, a difference emerged between children and adults in mathematical skill: children with low aperiodic activity exhibited higher mathematical abilities, while no relation was found in adults.
I thought this study is interesting and provides a novel point of view on how cognitive abilities contribute to mathematical ability and potential difference across individual. I have only few minor concerns which I detailed below.
Methods:
Please report the mean age of children as you did for the adult sample.
Results:
Line 439-441 This sentence breaks the age * aperiodic activity interaction description and should be move at the end of this paragraph. “Note that running the full model (Hayes’ Model) for aperiodic activity with working memory and number sense as parallel mediators, no mediation effects were found.”
Since verbal and visuospatial working memory were assessed with different tasks, i.e., the backwards digit recall task and the odd one out task, did the authors check for differential contributions of verbal and spatial WM in mathematical ability?
Discussion
I am not convinced that the paragraph described in the discussion about the role of three number sense scores as separate mediators on working memory (lines 481-487) is supported by the data described in the results section. Please provide a further explanation about that point.
Author Response
Dear dr Kaufmann, dr Zamarian, dr Wood and dr Klein,
We would like to thank you and the reviewers for the constructive comments on our submission of the manuscript entitled ‘Predicting Math Ability using Working Memory, Number Sense, and Neurophysiology in Children and Adults’. We appreciate the time and effort that went into the reviewing process. As instructed, below we have responded to the points made by Reviewers 1 and 2. Revisions that were made in the manuscript are indicated in italic in the response letter and marked in yellow in the revised manuscript.
We hope that you and the reviewers will find our response satisfactory and look forward to hearing from you about our submission.
Kind regards,
Nienke van Bueren
To start, we have made a couple of adjustments in the manuscript that were not mentioned by the two reviewers:
- We have corrected our preregistration on Open Science Framework with the following link: osf.io/37pyn.
- For 5 participants (all children), we noticed that the beta power calculated with the traditional method was run over a different electrode than Fz, and we therefore have corrected these values and the analyses (see Table 1, Figure 1b, and Table 2). Note that this slight change resulted in a significant Cmod (line 391), and we have therefore adjusted the manuscript accordingly at the end of paragraph ‘3.1. Moderated Mediation Models: The Parameterization Method and the Traditional Method’ in the Results section. However, the conclusions as presented in the discussion are not altered.
Reviewer 1
We would like to thank the reviewer for finding our study interesting and novel. The Reviewer has raised a few minor concerns which we have addressed below.
Please report the mean age of children as you did for the adult sample.
We have added the following part (line 208). “(M = 9 years, 7 months, SD = 0.64).
Results:
Line 439-441 This sentence breaks the age * aperiodic activity interaction description and should be move at the end of this paragraph. “Note that running the full model (Hayes’ Model) for aperiodic activity with working memory and number sense as parallel mediators, no mediation effects were found.”
We would like to thank the reviewer for this excellent suggestion of moving the referred sentence to not interfere with the previous mentioned interaction. We have therefore removed the sentence to the end of the first paragraph of section ‘3.3. Exploratory Analyses’ (line 444).
Since verbal and visuospatial working memory were assessed with different tasks, i.e., the backwards digit recall task and the odd one out task, did the authors check for differential contributions of verbal and spatial WM in mathematical ability?
Since our main aim was to focus on a generalized score of working memory, including verbal and visuospatial abilities, we have chosen to make a composite score of working memory by combining both scores. This was also done to not overly complicate the moderated mediation model and to not increase the complexity of our paper. To answer the reviewer’s question, we have checked for differential contributions of verbal and spatial working memory by running separate moderated mediation models with the tasks as parallel mediators. Visuospatial working memory was a significant predictor of mathematical ability (β = .53, SE = .12, p < .001) which was not the case for verbal working memory (β = .12, SE = .12, p = .31). When the outcome of the two tasks were run in separate models, then both visuospatial working memory (β = .57, SE = .12, p < .001) and verbal working memory were related to mathematical ability (β = .29, SE = .13, p = .03). This indicates a shared variance between the two tasks (i.e., working memory). Similar results were found when running the model for only multiplication (predominantly retrieval strategies) or only for subtraction problems (i.e, predominantly procedural strategies).
Discussion
I am not convinced that the paragraph described in the discussion about the role of three number sense scores as separate mediators on working memory (lines 481-487) is supported by the data described in the results section. Please provide a further explanation about that point.
We would like to thank the reviewer for recognizing this unclear description in our discussion section related to our results. To clarify this, we have revised the manuscript accordingly in the results section by adding an extra paragraph that discussed this extra (exploratory) analysis in more detail in paragraph “3.3. Exploratory analyses” (line 453). “Our results do not show a relation between number sense and mathematical ability, contrary to previous studies [7,10,11]. Also, the subscores of the individual number sense tasks (number line estimation task, symbolic and non-symbolic task) were weakly related to each other (all p < .045, all r > .20). Therefore, a new moderated mediation model was run with the individual number sense subscores included as parallel mediators (Hayes’s Model 59 including working memory as mediator). This was done to see whether a specific number sense skill (number line estimation, symbolic and non-symbolic comparison) was related to mathematical ability. Number line estimation did not predict mathematical ability (β = .17, SE = .11, p = .12), nor did symbolic comparison (β = .09, SE = .09, p = .35) or non-symbolic comparison (β = .14, SE = .12, p = .24). Additionally, rs-beta activity calculated with the traditional method did not predict number line estimation (β = .24, SE = .13, p = .06), symbolic comparison (β = .00, SE = .13, p = .96), or non-symbolic comparison (β = .04, SE = .14, p = .75). For beta activity calculated with the parameterization method a significant relation was found for number line estimation (β = -.32, SE = .12, p = .012), but not for symbolic comparison (β = .11, SE = .12, p = .38), or non-symbolic comparison (β = -.13, SE = .14, p = .34). These results indicate no relation between number sense and mathematical ability for different age groups, nor a relation with resting-state electrophysiology.”
Reviewer 2 Report
This article is very complex and technical. I will only make a few remarks of form and substance without overcoming my domains of competence.
With such a complex article, it is almost inevitable that it contains some errors. Even if the number of co-authors (= 5) should have contributed to reduce the number of these errors, I still want to quote one indisputable one. In Table 1, one of the t-values, which are not well aligned, seems to be grossly wrong: it is t = 29.28 for the variable "Accuracy OOO". This value on the one hand should be negative and on the other hand its order of magnitude is 7 or 8 (obviously, if the means and SDs are correct).
Regarding the assessment of arithmetic ability, I regret that you do not take into account wrong answers: children, and even adults, make mistakes! Moreover, not being familiar with Vos' test, I don't know if it has been sufficiently emphasized that the calculations must be done in the order in which they are presented. Otherwise, children (in particular) may skip difficult calculations or those they don't know by heart (in a declarative way).
If my reading of Figures S7 and S8 is sufficient, there seems to be little difference between addition/multiplication and subtraction/division. I wonder then if a comparison between multiplication (the most declarative operation) and subtraction (the most procedural operation), would not have led to a more differentiated result. Indeed, some small additions are certainly retrieved in memory and divisions allow, once and for all when not mixed with the other operations, to adopt a multiplicative strategy. This strategy consists, for example, in realizing that 27:3 activates the triplet (3, 9, 27) in declarative memory and allows the missing element of the triplet (namely 9) to be recovered as the answer to 27:3. We can thus see that additions and divisions may have introduced noise into the comparison.
As the statistician George Box, I think that all models are wrong. Moreover, when you analyze modeled data (or indirectly obtained data), the models does not take account of this. All your p, derived from the coefficients βs and SEs, seem correct, but the SE themselves do not take into account the additional uncertainty added by the imperfectness of the modeled data. Therefore it is not a surprise that many discrepancies appear in the article. Discrepancies with your own expectations (e.g., "Contrary to our expectations", line 458), or with the results of other research (e.g., exponent in Cellier et al.'s research: line 432-433).
Very punctual remarks: The last column in Table 2 seems unnecessary; lines 347 and 351: the repetition “average of the offset and exponent” also seems unnecessary.
Author Response
Dear dr Kaufmann, dr Zamarian, dr Wood and dr Klein,
We would like to thank you and the reviewers for the constructive comments on our submission of the manuscript entitled ‘Predicting Math Ability using Working Memory, Number Sense, and Neurophysiology in Children and Adults’. We appreciate the time and effort that went into the reviewing process. As instructed, below we have responded to the points made by Reviewers 1 and 2. Revisions that were made in the manuscript are indicated in italic in the response letter and marked in yellow in the revised manuscript.
We hope that you and the reviewers will find our response satisfactory and look forward to hearing from you about our submission.
Kind regards,
Nienke van Bueren
To start, we have made a couple of adjustments in the manuscript that were not mentioned by the two reviewers:
- We have corrected our preregistration on Open Science Framework with the following link: osf.io/37pyn.
- For 5 participants (all children), we noticed that the beta power calculated with the traditional method was run over a different electrode than Fz, and we therefore have corrected these values and the analyses (see Table 1, Figure 1b, and Table 2). Note that this slight change resulted in a significant Cmod (line 391), and we have therefore adjusted the manuscript accordingly at the end of paragraph ‘3.1. Moderated Mediation Models: The Parameterization Method and the Traditional Method’ in the Results section. However, the conclusions as presented in the discussion are not altered.
Reviewer 2
With such a complex article, it is almost inevitable that it contains some errors. Even if the number of co-authors (= 5) should have contributed to reduce the number of these errors, I still want to quote one indisputable one. In Table 1, one of the t-values, which are not well aligned, seems to be grossly wrong: it is t = 29.28 for the variable "Accuracy OOO". This value on the one hand should be negative and on the other hand its order of magnitude is 7 or 8 (obviously, if the means and SDs are correct).
We would like to thank the reviewer for pointing out this error. We have checked the mentioned variable (accuracy of the Odd One Out Task) accordingly and revised Table 1. In addition, we have aligned the table and also checked every other variable mentioned in Table 1 (Means, SDs and outcome of the t-test) to make sure there are no errors present.
Regarding the assessment of arithmetic ability, I regret that you do not take into account wrong answers: children, and even adults, make mistakes! Moreover, not being familiar with Vos' test, I don't know if it has been sufficiently emphasized that the calculations must be done in the order in which they are presented. Otherwise, children (in particular) may skip difficult calculations or those they don't know by heart (in a declarative way).
The Dutch speeded arithmetics test (TTR) is a validated math test that we administered according to the instructions of the test manual. The wrong answers were taken into account in the sense that they are not added to the total score (i.e. they are assigned a score of 0). Correct items were given a score of 1.
It is an interesting suggestion to analyze the wrong answers, but this is outside the scope of this paper, which is already heavy in the number of analyses, and complex, as the Reviewer rightly highlighted. Additionally, we have little information about the incorrect answers. For example, we do not know if the participants made a guess or seriously attempted to solve the problem. Also, participants did not make many mistakes, indicating that it might not be a useful measure.
Concerning the order: before the TTR was assessed, it was made clear to the participants that they solve the problems in the same order (from the top of the page; problem number 1 to the bottom of the page; problem number 40). To clarify this instruction given to the participants, we have added the following part in section “2.2.2. Arithmetic Ability” (line 263). “Participants were also instructed to solve the problems from the top of the page to the bottom of the page.”
If my reading of Figures S7 and S8 is sufficient, there seems to be little difference between addition/multiplication and subtraction/division. I wonder then if a comparison between multiplication (the most declarative operation) and subtraction (the most procedural operation), would not have led to a more differentiated result. Indeed, some small additions are certainly retrieved in memory and divisions allow, once and for all when not mixed with the other operations, to adopt a multiplicative strategy. This strategy consists, for example, in realizing that 27:3 activates the triplet (3, 9, 27) in declarative memory and allows the missing element of the triplet (namely 9) to be recovered as the answer to 27:3. We can thus see that additions and divisions may have introduced noise into the comparison.
We agree with the notion of the reviewer that procedural processing can be intertwined with declarative processing and vice versa, and that the data is not a 100% noise free due to the possibility that participants switch strategies during the task. Therefore, we have slightly adjusted our phrasing in the manuscript (line 362 and 364) to ‘predominantly retrieval strategies’ and ‘predominantly retrieval strategies’. As mentioned by the reviewer, we checked if the models in Figure S7 and S8 indeed relate to an increased differential effect when just comparing multiplication with subtraction. For comparisons, we have put Figure S7 and S8 below with the new models. When comparing panel C/D of Figure S7 with panel C/D of the new model that only contains multiplication and subtraction, there is little difference in the estimated of all paths. This is similar to panel C/D of Figure S8 with panel C/D of the new model.
Figure S7. Moderated mediation model for bilateral parietal-occipital theta activity (n = 105). (a) Location of the clustered electrodes for the bilateral parietal-occipital area PO9, P3, O1, PO10, P4, and O2. (b) Moderated mediation model that includes the bilateral parietal-occipital theta activity calculated with the parameterization method and mathematical ability. (c) Moderated mediation model that includes the bilateral parietal-occipital theta activity and addition and multiplication problems. (d) Moderated mediation model that includes bilateral parietal-occipital theta activity and subtraction and division problems.***p < .001.
Moderated mediation model for bilateral parietal-occipital theta activity (n = 105). (a) Location of the clustered electrodes for the bilateral parietal-occipital area PO9, P3, O1, PO10, P4, and O2. (b) Moderated mediation model that includes the bilateral parietal-occipital theta activity calculated with the parameterization method and mathematical ability. (c) Moderated mediation model that includes the bilateral parietal-occipital theta activity and multiplication problems. (d) Moderated mediation model that includes bilateral parietal-occipital theta activity and subtraction problems.***p < .001.
Figure S8. Moderated mediation model for bilateral parietal-occipital alpha activity (n = 105). (a) Location of the clustered electrodes for the bilateral parietal-occipital area PO9, P3, O1, PO10, P4, and O2. (b) Moderated mediation model that includes the bilateral parietal-occipital alpha activity calculated with the parameterization method and mathematical ability. (c) Moderated mediation model that includes the bilateral parietal-occipital alpha activity and mathematical retrieval (addition and multiplication). (d) Moderated mediation model that includes bilateral parietal-occipital alpha activity and mathematical procedural calculation (subtraction and division).***p < .001.
Moderated mediation model for bilateral parietal-occipital alpha activity (n = 105). (a) Location of the clustered electrodes for the bilateral parietal-occipital area PO9, P3, O1, PO10, P4, and O2. (b) Moderated mediation model that includes the bilateral parietal-occipital alpha activity calculated with the parameterization method and mathematical ability. (c) Moderated mediation model that includes the bilateral parietal-occipital alpha activity and mathematical retrieval (multiplication). (d) Moderated mediation model that includes bilateral parietal-occipital alpha activity and mathematical procedural calculation (subtraction).***p < .001.
As the statistician George Box, I think that all models are wrong. Moreover, when you analyze modeled data (or indirectly obtained data), the models does not take account of this. All your p, derived from the coefficients βs and SEs, seem correct, but the SE themselves do not take into account the additional uncertainty added by the imperfectness of the modeled data. Therefore it is not a surprise that many discrepancies appear in the article. Discrepancies with your own expectations (e.g., "Contrary to our expectations", line 458), or with the results of other research (e.g., exponent in Cellier et al.'s research: line 432-433).
We totally agree with the reviewer and George Box, but hope that the reviewer, as George Box stated, agree that some models might be useful. We agree with the reviewer that our models are not perfect, and that some noise might be introduced and that it will never represent reality fully. However, this is unfortunately the case in models from social, medical, and life sciences. We aimed to reduce the amount of noise in our moderated mediation models to be as small as possible by extending previous research. We disagree with the thought that the discrepancies in our manuscript relate primarily to the noise in our models. As explained in the discussion, we relate our contradictory findings from Cellier et al.’s research to a different clustering electrode. Contradictory findings of our own research relate to the different methods used of calculating rs-beta, in which the traditional method also measures aperiodic activity. In line with other behavioral studies, we show that higher performance on working memory tasks relates to higher mathematical skills (De Smedt et al., 2009; Friso-Van den Bos et al., 2013; Kroesbergen et al., 2009; Raghubar et al., 2010).
Very punctual remarks: The last column in Table 2 seems unnecessary; lines 347 and 351: the repetition “average of the offset and exponent” also seems unnecessary.
We would like to thank the reviewer for noticing these unnecessary additions. We have removed the last column from Table 2 to avoid redundancy. We have also removed the phrase “average of the offset and exponent” from lines 347-351 in the manuscript.